# A Cross-Sectional Study of Ocular Changes in Children and Adolescents with Diabetes Mellitus in Selected Health Facilities in Ghana

**DOI:** 10.3390/ijerph19095295

**Published:** 2022-04-27

**Authors:** Vera A. Essuman, Naa N. Tagoe, Akye Essuman, Benjamin Abaidoo, Josephine Akpalu, Harold A. Sackey, Charles F. Hayfron-Benjamin, George Asare, Albert G. B. Amoah, Thomas A. Ndanu, Imelda D. B. Ofori-Adjei, Nana A. Barnes, Benedicta L. Appiah-Thompson, Winfried M. Amoaku

**Affiliations:** 1Ophthalmology Unit, Department of Surgery, University of Ghana Medical School, University of Ghana, Accra P.O. Box GP 4236, Ghana; benjamin_abaidoo@yahoo.com; 2Eye Department, Korle Bu Teaching Hospital, Accra P.O. Box 77, Ghana; naanaamuahtagoe@gmail.com (N.N.T.); dziffabella@gmail.com (I.D.B.O.-A.); 3Department of Community Health, University of Ghana Medical School, University of Ghana, Accra P.O. Box GP 4236, Ghana; akye.essuman@gmail.com; 4Department of Medicine and Therapeutics, University of Ghana Medical School, University of Ghana, Accra P.O. Box GP 4236, Ghana; jakpalu@ug.edu.gh (J.A.); agbamoah@hotmail.com (A.G.B.A.); 5Department of Child Health, University of Ghana Medical School, University of Ghana, Accra P.O. Box GP 4236, Ghana; hasackey@ug.edu.gh; 6Department of Physiology, University of Ghana Medical School, University of Ghana, Accra P.O. Box GP 4236, Ghana; chayfron-benjamin@ug.edu.gh; 7Department of Anaesthesia, Korle Bu Teaching Hospital, Korle Bu, Accra P.O. Box GP 4236, Ghana; 8Chemical Pathology Unit, Department of Medical Laboratory Sciences, University of Ghana School of Basic and Allied Health Sciences, College of Health Sciences, Accra P.O. Box GP 4236, Ghana; george.asare@gmail.com; 9National Diabetes Management and Research Centre (NDMRC), Korle Bu Teaching Hospital, Accra P.O. Box GP 4236, Ghana; 10Department of Preventive & Community Dentistry, University of Ghana Dental School, University of Ghana, Accra P.O. Box GP 4236, Ghana; revtomdata@gmail.com; 11Santa Rosa Community Health, Vista Clinic 3569 Round Barn Circle, Santa Rosa, CA 95403, USA; nana_barnes@yahoo.com; 12Eye Department, Cape-Coast Teaching Hospital, Cape Coast CT 1363, Ghana; bappiahthompson@gmail.com; 13Ophthalmology and Visual Sciences (DCN), University Hospital, QMC Nottingham, Nottingham NG7 2UH, UK; winfried.amoaku@nottingham.ac.uk

**Keywords:** diabetes mellitus, microvascular, ocular complications, retinopathy

## Abstract

Background: The main objective of the study was to determine the prevalence of diabetic retinopathy (DR), other diabetes-related ocular changes (e.g., cataracts, corneal ulceration), and non-diabetic ocular disease in Ghanaian children and adolescents. The second objective was to evaluate the relationship between these conditions and age at diagnosis, current age, diabetes mellitus (DM) duration, and participant’s sex. Methods: A cross-sectional study, undertaken by a multidisciplinary team, included a cohort of children and adolescents (4–19 years) with DM recruited from selected health facilities in Ghana, from March 2016 to September 2019, after written informed consent or assent. The cohort will be followed up for 3 years to determine the natural course of the ocular changes, reported later. Participants were examined for all microvascular and macrovascular complications, non-diabetic ocular disease, anthropometric measurements, laboratory characteristics and quality of life issues. Full ocular examination was also undertaken. Statistical Package for Social Sciences (SPSS Version 25.0) was used for the data analysis. Continuous and categorical variables were presented as mean and standard deviation (SD), median (interquartile range) and as percentages (%), respectively. T-test and Mann–Whitney U test were used in establishing associations. Results: A total of 58 participants were recruited. DR was detected in only 1 out of 58 (1.7%) participants at baseline. Cataracts were the most common ocular finding, detected in 42 (72%) at baseline. Other anterior segment changes observed included blepharitis 46 (79.3%) and tear film instability 38 (65.5%). There was a significant positive association between duration of the DM and the risk of cataract (*p* = 0.027). Participants’ age at diagnosis was significantly associated with the presence of prominent corneal nerves (*p* = 0.004). Conclusions: DR was uncommon in this cohort of young persons with DM in Ghana. Cataracts, blepharitis and refractive errors were ocular changes commonly observed. All young persons with diabetes should undergo regular eye examination in all clinics where follow-up care is provided.

## 1. Introduction

Diabetes mellitus (DM), is a chronic systemic metabolic disease characterized by hyperglycaemia, associated with microvascular and macrovascular complications, which lead to significant morbidity, especially kidney, eye, heart and cerebrovascular disease, and premature death [1,2]. The worldwide prevalence of DM continues to increase [3,4]. Traditionally, type 1 DM (T1DM) is more common in young people; however, type 2 DM (T2DM) in young people has also been increasing worldwide. [4]. Although data exists on DM prevalence in sub-Saharan Africa (SSA), almost all pertain to adults [5,6]. The limited data (most of which is old) [7,8] on DM prevalence in SSA children ranges from 10.1 per 100,000 to 0.33 per 1000 in a Nigerian hospital for persons below 15 years [7,8]. In Ghana, there is a dearth of information on the incidence and prevalence of DM in children and adolescents. Limited clinic-based data in Ghana suggest that the prevalence of childhood and adolescent diabetes is on the increase, similar to the trend in ethnic minority population in western countries [9]. A retrospective review based on data from a tertiary hospital in Ghana showed that the number of individuals diagnosed with T1DM decreased among the 10–19 years age group during the 1992–1994 period, followed by a progressive increase within the same age group during the subsequent period (from 35.4% in 1995–1997 to 63.2% in 2016–2018) [10]. There was, however, a decline in the proportion of children 0–9 years of age diagnosed during the study period (from 5.1% in 1992–1994 to 3.6% in 2016–2018) [10].

It is well recognized that children and adolescents with type 1 DM are faced with the threat of acute complications of hypoglycemia and ketoacidosis on a day-to-day basis [11]. However, it is the microvascular and macrovascular complications of the disease, in the long term, which place them at greatest risk for serious morbidity and early mortality [11,12].

DM can affect any ocular tissue, although the retinal microvascular changes are thought to be most important, especially in older persons [1,2,12]. Younger persons with DM, however, have increased risks of developing significant non-vascular diabetic eye morbidities, including neurotrophic keratitis, cataracts, and cranial nerve palsies, as well as refractive abnormalities [2,13]. Tear film abnormalities, and corneal nerve changes in young persons have also been recently described [14,15]. Diabetic retinopathy (DR) is one of the long-term microvascular complications of DM in children and leads to a debilitating effect on vision and significant visual loss in some cases [1,2]. The available literature suggests that DR onset is variable and may be present at the time of DM diagnosis [12]. This DR risk variability is corroborated by a population study from Australia where 24% of children and adolescents with type 1 DM developed retinopathy after 6 years [16]. Furthermore, the incidence of DR has been reported as dependent on age of diagnosis and duration [16]. A nationwide prospective cohort study of children and adolescents with type 1 DM from Denmark reported the prevalence of any level of DR as 17.7% in 12–15 years, 45.4% in 16–20 years age groups, increasing to 67.6% in those older than 20 years [17]. However, recent reports suggest significant reductions in DR prevalence, which may be variable, and that compliance with DR screening is variable in different countries [18,19]. DR was significantly associated with poor long-term glycemic control (as measured by glycated hemoglobin [HbA1c] levels) and DM duration [17].

Despite the recognition of DM predisposition to the development of generalized microangiopathy, and macrovascular disease in children and adolescents [16,20,21], it is also known that clinical characteristics of T1DM and T2DM in people from SSA can differ somewhat from that seen in typical European populations [9]. A recent report from Egypt reported a 10.4% prevalence of DR [22]. To the best of our knowledge, there is limited or no data on the occurrence of ocular disease in children with diabetes in SSA, including Ghana.

In this cross-sectional study we provide a relatively comprehensive ocular assessment of Ghanaian children and adolescents with diabetes from two tertiary healthcare facilities. In particular, the study sought to determine the hospital prevalence of ocular disease in children and adolescents with diabetes in Ghana, including DR, other diabetes-related ocular changes (cataracts, corneal ulceration) and non-diabetic ocular disease. It further evaluated the relationship between these conditions and age at DM diagnosis, age at study enrolment, duration of DM, and the sex of the participants.

## 2. Materials and Methods

The study was conducted in 2 phases comprising an initial cross-sectional study to determine the hospital prevalence and characteristics of ocular disease in children and adolescents with diabetes in Ghana, including DR, other diabetes-related ocular changes (cataracts, corneal ulceration) and non-diabetic ocular disease among participants consecutively attending the study hospitals over a 3-year period. The second phase was a longitudinal study in which participants from the first phase were followed up annually for up to 3 years to determine the natural history of the studied conditions. We report on the cross-sectional arm of the study which enrolled children and adolescents aged 4–19 years with DM. A sample size of at least 55 participants were required to meet the primary outcomes, with a power of 80% at a level of significance of 0.05 [23]. Participants were consecutively recruited from the out-patient clinics of the Departments of Child Health, Medicine and Therapeutics, Family Medicine, Ophthalmology, and the National Diabetes Management and Research Centre, all at the Korle Bu Teaching Hospital (KBTH), Accra, and Cape-Coast Teaching Hospital (CCTH) in the Central Region of Ghana from March 2016 to September 2019. Written informed consent or assent, as appropriate, was obtained from each participant or parent/guardian. Institutional approval from the Ethical and Protocol Review Committee (EPRC) of the University of Ghana Medical School (Protocol Identification Number: MS-Et/M.12-P4.5/2013-2014) was granted. Ethics approval was also obtained from all the participating institutions. The study complied with the tenets of the Helsinki Declaration of 2014. The full study protocol was described in the study design and rationale manuscript [23].

Children and adolescents aged between 4 to 19 years, diagnosed with DM (by their attending physicians) and attendants at the KBTH or the CCTH who gave consent, assent or whose parents/guardians provided consent were included. Children and adolescents aged 4–19 years diagnosed with DM and concomitant chronic disease, psychiatric disease, or coexisting acute disease during the study period were excluded.

As described in our earlier publication [23], diagnosis of DM was made in patients with classic symptoms and/or fasting plasma glucose of >7 mmol/L, random plasma glucose of >11.1 (as diagnosed by their attending physicians). DM was further characterized as type 1 DM, by the presence of GAD 65 antibodies, islet cell antibodies, insulinopenia or low levels of C-peptide (stimulated C peptide values of <0.6 pmol/mL), or type 2 DM indicated by a negative test for type 1 DM-specific antibodies i.e., negative GAD-65 antibodies, negative islet cell antibodies, in association with elevated fasting insulin or C-peptide (stimulated C-peptide assay of ≥0.6 pmol/mL), or the presence of acanthosis nigricans. Other types of diabetes such as Maturity-onset Diabetes in Youth (MODY) or Atypical Diabetes in childhood and adolescents were not characterized.

Demographic information, medical history, anthropometric measurements, and laboratory characteristics were recorded using a predesigned questionnaire. All the participants received a full ocular examination by ophthalmologists, as well as examination by the multidisciplinary team of physicians and pediatrician, and investigations conducted by laboratory scientists.

Ocular examinations included visual acuity (VA) assessment using the appropriate LogMAR test type for age. The visual acuity was further categorized as normal, mild, moderate, severe vision reduction (or visual impairment) and blindness as categorized in the ICD-11 classification [24]. Tear Break Up Time (TBUT) was assessed using standardized methods as described by Kesarwani et al. [25] where TBUT test was performed by staining the tear film using a fluorescein strip without using topical anesthesia and asking the subjects to blink several times for few seconds to encourage its distribution. The tear film was observed using a cobalt blue filter without artificially holding the lids open. The interval between the last blink and the appearance of the first corneal dry spot in the stained tear film was measured. The procedure was repeated 3 times and the mean value recorded. A TBUT was considered abnormal when the value was <10 s. The anterior ocular segment as well as anterior vitreous were assessed using slit lamp biomicroscopy (with particular attention to corneal innervation. Confocal microscopy was not available). Dilated fundus biomicroscopy using 78 D or 90 D lens with the slit lamp was used to assess the vitreous and fundus, complemented with binocular indirect ophthalmoscopy with a 20 D, or 28 D lens. Ocular ultrasound (B mode) examination of the posterior segment of the eye with a Tomey UD 1000 Model (Tomey, Germany) was conducted to assess the vitreous, in particular, where media opacity precluded fundus visualization. Ocular fundus photography acquiring 4-field stereo color photos from each eye with a VISUSCOUT 100 portable retina camera system, (Jena, Germany), was conducted. All retinal images were graded independently (according to the ETDRS) by WMA. Optical Coherence Tomography (OCT) was performed in participants who demonstrated macular and optic nerve pathology.

In brief, a full physical examination of each participant was undertaken by the multidisciplinary team of diabetes, and family physicians, and pediatrician in the study team. Anthropometric measurements of height and weight were taken, Body Mass Index (BMI) calculated in kilograms per square meter, and BMI—age-sex z scores and percentiles determined using CDC reference standards and AnthroPlus 2005 software. Laboratory investigations to further categorize the type of DM and assess the level of DM control were undertaken [23].

Before the commencement of the study, the study team visited the various study centres to explain the study procedures and the risk and benefits associated with the study to the participants in English and their local dialect. Executives and members of the Diabetes Youth Care Ghana (a public organization for children and adolescents with diabetes in Ghana) were actively involved in the recruitment of participants. Written informed consent with signature or inked fingerprint were obtained to show the commitment of participants to voluntarily participate in the study.

The primary outcome was the prevalence of DR at baseline in this population.

Secondary outcomes included the prevalence of other ocular changes of diabetes (cataracts, corneal ulceration) and non-diabetic ocular disease in children and adolescents at baseline.

The Statistical Package for Social Sciences (SPSS Version 25.0) was used for data capture and analysis. A test of normality was conducted for variables including age of participants, age at diagnosis, BMI and, duration of DM using the Kolmogorov-Smirnov test, with *p*-values less than 0.05 considered as not normally distributed. Continuous and categorical variables were presented as mean and standard deviation (SD), median (interquartile range) and as percentages (%), respectively. T-test was used in establishing associations between various ocular conditions and current mean age and mean age at diagnosis. Mann–Whitney U test of independence was used to evaluate the association between the various ocular parameters and demographic and clinical characteristics. Significance was set at α = 0.05.

## 3. Results

Fifty-eight participants were recruited from the two centres over the 3-year period, including 45 (77.6%) females, as reported in our published protocol [23]. From the test of normality, age, age at diagnosis, BMI, and duration of DM were not normally distributed (*p*-values < 0.05). Participants’ ages ranged from 5 to 19 years with a median age of 15.0 (13.0–16.0) years at baseline. The median age at DM diagnosis was 11.0 (9.0–13.3) years. The median durations between time of diagnosis of DM and time of the first eye examination was 3.0 (1.0–6.3) years. Fifty-six (96.6%) participants had type 1 DM. The mean HBA1c was 10.1 (2.9%). The mean BMI was 20.7 (4.9) kg/m^2^. Insulin (53, 91.4%) was the main treatment for DM in this cohort. Most of the participants on insulin used the biphasic human insulin (mixtard) type (48, 82.8%). Three participants were on oral hypoglycemic treatment (metformin) and two (3.4%) had a combination of insulin and oral hypoglycemic treatment.

### 3.1. Visual Impairment in Children and Adolescents with DM

Table 1a summarizes the presenting visual acuity/impairment in children and adolescents with DM. The majority of the participants had normal vision and none was blind or had severe visual impairment. Visual impairment (VI) was noted in the right eye of eight participants (13.7%), and in the left eye of eight participants (13.7%). Overall, VI occurred in 16 (13.8%) out of 116 eyes in this study. The VI was mild in most cases, due to refractive error, juvenile glaucoma, and cataract. Two participants who had juvenile glaucoma had moderate VI in at least one eye; one had mild VI in the left eye (Table 1a). Vision in all eyes with mild VI and one eye with moderate VI were correctable to normal vision with spectacles, with only one person remaining with VI (moderate) with best correction (Table 1a,b).

### 3.2. Diabetic and Non-Diabetic Ocular Changes in Children and Adolescents with DM

DR was detected in both eyes of 1 participant out of 58 at baseline examination, equating to a prevalence of DR at baseline of 1.7%. This participant had a moderate non-proliferative diabetic retinopathy (NPDR) presenting with dot and blot haemorrhages, macular exudates and cotton wool spot in both eyes. In addition, she had bilateral cataracts (cortical and posterior subcapsular), prominent corneal nerves, pseudogerontoxon and blepharitis (mixed).

At enrolment, she was 17 years old with type 1 diabetes of 7 years duration, HbA1c was 11.8%, blood pressure (BP) standing was 131/98 mmHg, lying 135/86 mmHg, pulse 86 beats per minute, and BMI of 21.8 kg/m^2^ (50–75th percentile).

Table 2 summarizes the other ocular and adnexal findings, and other diagnoses in participants at baseline. Cataracts were observed in 42 (71.2%) of the participants. Among the different types of cataracts encountered, the posterior subcapsular (40, 69.0%) and the cortical (19, 32.8%) types were the most common types (*p* < 0.001). Blepharitis, and abnormal TBUT were very common in both eyes in this DM cohort. Other anterior segment and adnexal manifestations included prominent corneal nerves, superficial corneal vascularization, peripheral corneal scarring, cornea shield ulcers in both eyes from severe vernal keratoconjunctivitis, and posterior embryotoxon. Other features detected in this cohort included early pterygium, lid discharge, myasthenic lid fatigability, ptosis, chalazion, subconjunctival melanosis, subtarsal papillae, tear film debris, and lymphoid follicular hypertrophy.

The commonest posterior segment findings encountered were vitreal syneresis in 21 (36.2%) and optic neuropathy in 6 (10.3%) participants. In the six participants with optic neuropathy, four had juvenile glaucoma, one retinitis pigmentosa, and the other had mild optic atrophy of indeterminate cause. Other posterior segment findings included lattice degeneration, tortuous retinal arterioles, chorioretinal scar, congenital hyperplasia of the retinal pigment epithelium (CHRPE), unmasking of peripheral choroidal vessels due to retinal atrophy, tortuous engorged retinal veins, and tortuous vessels on disc (both probably congenital) (Table 2). However, these findings were infrequent.

Blepharitis was significantly higher in proportion than the rest of the adnexal conditions encountered (*p* < 0.001). Ocular innervation abnormalities are summarized in Table 2. Reduction in touch sensitivity was detected in one participant. No other abnormality was found to explain this finding. Cranial nerve (CN) I dysfunction was found in one person, and CN IV in another; however, the causes could not be determined. In one other participant, CN III weakness was attributed to a diagnosis of myasthenia gravis, with the addition of lid fatigability and ptosis. She was receiving treatment for myasthenia gravis at the time of examination.

### 3.3. Association between Ocular Findings and Age at Diagnosis, Current Age, Duration of DM and Sex of Children and Adolescents with DM

There was a significant positive association between duration of the DM and the risk of cataract (*p* = 0.027). The presence of prominent corneal nerves on biomicroscopy was significantly associated with age at diagnosis (*p* = 0.004) but not with current age or duration of DM (*p* > 0.05). No significant association was observed between syneresis with current age, age at diagnosis, or duration of DM (*p* > 0.05) (Table 3). Other identified ocular conditions were too infrequent to allow for any test of association. There was no significant association between sex and any of the ocular conditions observed in the children with DM (*p* > 0.05) (Table 4).

### 3.4. Summary of Results

Fifty-eight (58) children and adolescents were recruited. A female preponderance (45, 77.6%) was observed. Median age of participants, median age at diagnosis, and median duration of DM were 15.0 (13.0–16.0) years, 11.0 (9.0–13.3) years and 3.0 (1.0–6.3) years, respectively. Overall, VI occurred in 16 (13.8%) out of 116 eyes in this study. Vision in all eyes with mild VI and one eye with moderate VI were correctable to normal vision with spectacles, with only one person remaining with VI (moderate) with best correction. Cataracts were observed in 42 (71.2%) of the participants. Among the different types of cataracts encountered, the posterior subcapsular (40, 69.0%) and the cortical (19, 32.8%) types were the most common types. There was a significant association between duration of the DM and the risk of cataract (*p* = 0.027), and presence of prominent corneal nerves was significantly associated with age at diagnosis (*p* = 0.004). Other anterior segment changes observed included blepharitis, 46 (79.3%) and tear film instability, 38 (65.5%). DR was detected in only one (1.7%) out of 58 participants at baseline.

## 4. Discussion

It is generally acknowledged that both T1 and T2 DM in SSA are increasing [3]. However, there is a paucity of data on the ophthalmological changes in DM in this sub region. To the best of our knowledge, this study provides the first comprehensive data on the ocular changes in children and adolescents with DM in SSA, and in particular, Ghana, and serves as the basis for further detailed studies. Type 2 DM was very uncommon in our cohort and significantly less than what is reported elsewhere [26,27].

The prevalence of VI in this current study cohort was low, and was associated with refractive error, juvenile glaucoma, and cataract. Significant ocular surface abnormalities in children and adolescents with DM were observed in this study. Such tear film abnormalities in adults with diabetes have been confirmed in a meta-analysis by Yoo & Oh [28]. Inanc et al. [29] showed that reduction in TBUT in children with type 1 DM may be asymptomatic and was significantly associated with the duration of DM. Although we did not evaluate tear film osmolarity, Gunay et al. [14] suggested that ocular surface abnormalities in diabetic children may be due to increased tear film osmolarity. Older age at diagnosis, but not duration of DM, seems to be related to TBUT abnormalities. Frequency of blepharitis was related to the duration as well as the age of diagnosis of DM in this series. Similarly, increased meibomian gland dysfunction has been described in adult patients with diabetes, and may be asymptomatic [28,30]. However, we did not evaluate meibomian gland function directly in the current study. The finding of vernal conjunctivitis in one participant is probably incidental although this may have aggravated the ocular surface problems in this particular individual.

Reduced corneal sensitivity and loss of corneal sub-basal plexus have previously been described in type 1 DM [15]. Similarly, Nelson et al. [31] previously reported that diabetic corneal nerve changes are present in a quarter of children at diagnosis, and approximately half five years from diagnosis. Only one person demonstrated reduced corneal sensitivity. We were unable to quantify corneal nerve changes or measure retinal thickness changes in this cohort because of lack of equipment. It is possible that residual nerve fibres have compensated for lost fibres, thus increasing their prominence. It is planned that future studies will investigate these parameters in a Ghanaian cohort of type 1 DM.

Porter et al. [27], in their retrospective study of 343 patients (aged <21 years), reported an overall DR prevalence of 3.8%, including 3.4% in type 1 DM with duration of 10.4 + 3.8 years, and 6% in type 2 DM with duration of 2.7 + 2.5 years, with one case of PDR and moderate NPDR in three.

Previous studies have reported prevalence rates of cataracts in childhood DM, varying from 0.7 to 3.4% [13,32,33], with varying morphologies [13,32]. Significantly, all these studies were based on cohorts from developed countries. In our series, cataracts were a common finding, observed at a mean age of 14.6 years, compared to 13.6 years in the Geloneck study [32], after a mean interval of 3 years, compared to 5.3 years reported in the Geloneck study [32]. Longer duration of the DM was related to higher probability of developing cataracts in our series, similar to findings from previous reports [13,32]. The higher cataract prevalence of 72.4% in this study, which is much higher than previously reported, may be related to the local environment: temperature and potential risk of dehydration. The commonest type of cataract encountered in this cohort is the posterior subcapsular cataract, followed by cortical type. The snowflake subcapsular type of cataract has been described as the typical cataract associated with diabetes mellitus [34], but this study did not support that observation in our cohort, because that morphology was infrequently seen. The presence of cataract was not associated with poor vision at baseline because the degree of the cataract was variable. However, there is a need for close monitoring for progression because with progression, the cataracts, unfortunately, will lead to VI in a significant number of children and adolescents with DM. Prevalence of progressive cataracts is likely to interfere with subsequent fundus assessment for presence of DR, in addition to visual impairment, unless adequately managed. Furthermore, refractive errors (not associated with cataract) were observed in 11% of participants in Geloneck’s series [32], compared to 10.3% in ours. Strabismus and ocular nerve palsies were uncommon in our series compared to Geloneck’s series [32]. The association of juvenile glaucoma in this series was not statistically significant. This current study demonstrated many non-diabetic ocular changes among children with diabetes mellitus, especially cataract. Anecdotally, this high prevalence of cataract is higher than that seen among children who attend the largest tertiary paediatric eye unit in the country, the Ophthalmology Department, Korle Bu Teaching Hospital, which is one of the sites for this current study. There is no population-based study on childhood eye diseases in Ghana for comparison. Future population-based studies will be helpful.

This current study demonstrated a low prevalence of DR at baseline of 1.7% in our hospital cohort. This was a moderate NPDR. Potential confounders for such low prevalence include an under-reporting, or non-attendance of children with ocular morbidities, although we believe these are unlikely. Older studies from Tanzania reported a higher prevalence of background DR (BGDR), which was unusually much higher in pre-pubertal (25.8%) compared to pubertal (8.6%) children [35]. A higher prevalence of 14%, was also reported in the earlier study of Neuhann et al. [36] of type 1 DM. In a retrospective study from Haiti, cataracts were observed in 16%, and DR in 18%, with mean diabetes duration of 3 years [37]. However, this included participants as old as 26 years. The low prevalence of DR in our cohort may be related to the small size cohort, as well as the average short duration of DM. This finding is, however, not dissimilar to that of recent reports described elsewhere. No cases of DR were reported in the recent study by Abdella et al. [19] in the Canadian screening program, or in a US series published by Geloneck [32]. Older studies elsewhere, including Australia and Europe, reported DR rates as 25% in Australia, 17.7% in 12–15-year-olds, and 45% in 16–20-year-olds in Denmark [16,17]. However, in a US study, DR prevalence in persons under 21 years old were 20.1% and 7.2% in type 1 and type 2 DM, respectively [38]. A recent report from the UK showed that DR was uncommon (compared to older populations) and detectable only in 20% and 11% at the age of 12 and 13 years, respectively, in children who were diagnosed with diabetes at the age of 2 years or younger [18]. These DR rates were much lower in children diagnosed with diabetes between 2 and 12 years, at 8% and 2%, respectively. The Scanlon study confirmed that DR was likely to occur in children who develop DM at an earlier age than others, and that it is usually of a mild severity, with an incidence of 4.7% [18]. Ng et al. [39] reported a DR prevalence of 11% (consisting of 44% BGDR, and 56% pre-proliferative DR) in persons aged 12–19 years in northwest UK. This finding supports the UK recommendation of annual DR screening in children with DM from the age of 12 years and older [40]. The UK recommendation is contrary to the American Academy of Ophthalmology (AAO) recommendation, which recommends screening for children with diabetes in the USA 3–5 years after diagnosis of DM [41]. The Canadian Diabetes Association recommends commencement of screening at 15 years, 5 years after diagnosis of type 1, and at diagnosis of type 2 DM [42].

There are currently no existing guidelines for DR screening in children with DM in Ghana. Our findings support a delayed screening for DR until the age of 12 years, in line with The UK National Institute for Health and Care Excellence (NICE) recommendations [40]. However, in light of other significant ocular associations, including cataract, abnormal TBUT, and blepharitis, a different approach to managing pediatric diabetes in Ghana, and likely elsewhere in SSA, may be required. Findings from the longitudinal study involving this cohort of participants would offer insight into progression of some of the ocular changes encountered and help plan for screening and treatment interventions for other children and adolescents with diabetes mellitus in other parts of the country who are not currently on any routine screening program. An adequately resourced 3–5-year prospective multicentre evaluation of children and adolescents with diabetes in Ghana would be useful in guiding future national policy and management and may influence management of children and adolescents with DM in sub-Saharan Africa as well.

We recommend annual ophthalmic examination in all young persons with DM as part of their annual diabetes review, specifically looking out for potentially blinding diseases, such as cataract, and monitoring for progression. The inception of such an examination will allow early detection of other ocular diseases in younger children with diabetes and inform the basis for more formal screening programs in the future. Early and prompt therapeutic interventions should be put in place to manage these ocular diseases, especially cataract, glaucoma, and refractive errors, which are major causes of childhood blindness and visual impairment in children and adolescents. There is a need for education of caregivers, guardians, and patients with diabetes on the importance of screening for ocular complications.

The limitations of this study include the relatively small sample size, the inclusion of only hospitals for recruitment of participants, the absence of a control group, the lack of objective assessment of corneal innervation, and morphological assessments of the macula (retinal and choroidal thickness) with OCT. Though visual acuity is the conventional/standard mode of assessing visual function, some patients with cataract may have good visual acuity but may complain of impaired vision, necessitating the use of other visual function assessments such as contrast sensitivity. These assessments were planned but excluded due to inadequate resources. It is hoped that these assessments will be included in future studies for completeness. Education on diabetes for different stakeholders will improve uptake of research and clinical care in the susceptible population in the future.

## 5. Conclusions

DR was uncommon in this hospital cohort of children and adolescents with diabetes in Ghana. Other ocular changes including cataracts, blepharitis, and refractive errors were commonly observed. All young persons with diabetes mellitus should undergo regular eye examination in all clinics where follow-up care is provided.

## Figures and Tables

**Table 1 ijerph-19-05295-t001:** (**a**) Distribution of subjects according to presenting visual acuity/impairment in children and adolescents with diabetes mellitus, N = 58. (**b**) Distribution of subjects according to best-corrected visual acuity/impairment in children and adolescents with diabetes mellitus N = 58.

(a)		
Visual Impairment Categorisation	Right Eye N (%)	Left Eye N (%)
Normal vision [VA 6/12 or better (Snellen Equivalent)]	50 (86.2)	50 (86.2)
Mild Visual Impairment [VA worse than 6/12 to 6/18 (Snellen equivalent)]	6 (10.4)	7 (12.1)
Moderate Visual Impairment [VA worse than 6/18 to 6/60 (Snellen Equivalent)]	2 (3.4)	1 (1.7)
Severe Visual Impairment [VA worse than 6/60 to 3/60 (Snellen Equivalent)]	0.0	0.0
Blindness [VA worse than 3/60 (Snellen Equivalent)]	0.0	0.0
Total	58 (100.0)	58 (100.0)
(**b**)		
**Visual Impairment Categorisation**	**Right Eye N (%)**	**Left Eye N (%)**
Normal vision [VA 6/12 or better (Snellen Equivalent)]	57(98.3)	57 (98.3)
Mild Visual Impairment [VA worse than 6/12 to 6/18 (Snellen equivalent)]	0.0	0.0
Moderate Visual Impairment [VA worse than 6/18 to 6/60 (Snellen Equivalent)]	1 (1.7)	1 (1.7)
Severe Visual Impairment [VA worse than 6/60 to 3/60 (Snellen Equivalent)]	0.0	0.0
Blindness [VA worse than 3/60 (Snellen Equivalent)]	0.0	0.0
Total	58 (100.0)	58 (100.0)

Visual impairment categorisation as per ICD 11 (2018). VA = Visual Acuity, N = Number.

**Table 2 ijerph-19-05295-t002:** Distribution of subjects according to prevalence of other ocular conditions at baseline in 58 children and adolescents with diabetes mellitus.

Ocular Manifestations	Right Eye: N (%)	Left Eye: N (%)
**Adnexal**		
Dry eyes	10 (17.2)	10 (17.2)
Tear break up time (TBUT):		
*Normal*	20 (34.5)	21 (36.2)
*Abnormal*	38 (65.5)	37 (63.8)
Blepharitis	46 (79.3)	46 (79.3)
Allergic conjunctivitis	5 (8.6)	5 (8.6)
External hordeolum	2 (3.4)	2 (3.4)
Other adnexal manifestations	9 (15.5)	10 (17.2)
**Anterior Segment**		
Corneal nerves prominence	12 (20.7)	10 (17.2)
Pseudogerontoxon	5 (8.6)	5 (8.6)
Corneal pigmentation	2 (3.4)	1 (1.7)
Cataract		
*Present*	42 (72.4)	42 (72.4)
*Absent*	16 (27.6)	16 (27.6)
Types of Cataracts		
*Cortical*	19 (32.8)	19 (32.8)
*Nuclear*	4 (6.9)	4 (6.9)
*Sutural*	9 (15.5)	9 (15.5)
*Posterior subcapsular*	40 (69.0)	41 (70.7)
*Snowflake subcapsular*	9 (15.5)	9 (15.5)
*Christmas tree*	3 (5.2)	3 (5.2)
Other Anterior Segment Manifestations	4 (6.9)	4 (6.9)
**Posterior segment**		
Vitreous cells	1 (1.7)	1 (1.7)
Cotton wool spot (DR)	1 (1.7)	1 (1.7)
Dot and blot haemorrhages (DR)	1 (1.7)	1 (1.7)
Macular exudates (DR)	1 (1.7)	1 (1.7)
Optic neuropathy (include glaucoma)	6 (10.3)	6 (10.3)
Vitreous Syneresis	21 (36.2)	21(36.2)
Other posterior segment manifestations	7 (12.1)	7 (12.1)
**Other conditions**		
Refractive error	6 (10.3)	6 (10.3)
Juvenile Glaucoma	4 (6.9)	4 (6.9)
Pigmentary Maculopathy (indeterminate)	1 (1.7)	1 (1.7)
Myaesthenia Gravis	1 (1.7)	1 (1.7)
Retinitis Pigmentosa	1 (1.7)	1 (1.7)
Severe Vernal Keratoconjunctivitis	1 (1.7)	1 (1.7)
Nystagmus	1(1.7)	1(1.7)
**Impaired cranial nerve function**		
I	1 (1.7)	1 (1.7)
II	6 (10.3)	6 (10.3)
III	1 (1.7)	1 (1.7)
IV	-	1 (1.7)
V	1 (1.7)	1 (1.7)

Tear Break Up Time = TBUT, N = Number, Diabetic Retinopathy = DR.

**Table 3 ijerph-19-05295-t003:** Distribution of subjects according to associations between various ocular findings and current age, age at diagnosis, and duration of diabetes in children and adolescents with diabetes mellitus.

Ocular Condition	Age (Years) Mean (SD)	Age at Diagnosis (Years) Mean (SD)	Duration of DM (Months) Mean (SD)/Median
Cataract			
*Present*	14.6 (2.3)	11.1 (3.1)	26.2 (30.0)/12.0
*Absent*	14.1 (3.7)	9.6 (3.8)	5.9 (3.6)/5.0
*p-value*	*0.511*	*0.109*	***0.027* ***
Blepharitis			
*Present*	14.7 (2.4)	11.6 (2.8)	14.9 (22.4)/7.0
*Absent*	13.4 (3.5)	8.1 (3.5)	36.5 (33.1)/36.0
*p-value*	*0.075*	***<0.001* ***	***0.023* ***
TBUT			
*Normal*	14.7 (2.5)	10.7 (3.4)	25.4 (30.9)/12.0
*Abnormal*	14.2 (3.1)	10.7 (3.4)	12.6 (19.9)/6.0
*p-value*	*0.511*	*0.994*	*0.071*
Corneal nerve			
*Present*	15.0 (2.4)	13.00 (2.7)	17.6 (24.4)/5.0
*Absent*	14.4 (2.9)	10.0 (3.3)	21.2 (27.9)/9.0
*p-value*	*0.455*	***0.004* ***	*0.595*
Syneresis			
*Present*	14.1 (2.4)	11.1 (2.7)	16.1 (24.4)/7.0
*Absent*	14.7 (2.9)	10.5 (3.3)	22.8 (27.9)/10.0
*p-value*	0.415	0.453	0.250

DM = Diabetes Mellitus, SD = Standard Deviation, TBUT = Tear Break Up Time. * = Statistically significant values.

**Table 4 ijerph-19-05295-t004:** Distribution of subjects according to association between ocular changes and sex of children and adolescents with diabetes mellitus, N = 58.

Ocular Conditions	Male N (%)	Female N (%)	*p*-Value N (%)
Cataract			
*Present*	8 (61.5)	34 (75.6)	0.319
*Absent*	5 (38.5)	11 (24.4)	
Blepharitis			
*Present*	10 (71.4)	36 (76.1)	0.612
*Absent*	4 (28.6)	11 (23.9)	
Corneal nerve			
*Present*	3 (23.1)	10 (21.7)	0.918
*Absent*	10 (76.9)	36 (78.3)	
TBUT			
*Normal*	8 (61.5)	25 (55.6)	0.701
*Abnormal*	5 (38.5)	20(44.4)	
Syneresis			
*Present*	6 (46.2)	15 (32.6)	0.368
*Absent*	7 (53.8)	31 (67.4)	

N = Number, TBUT = Tear Break Up Time.

## Data Availability

The datasets used and/or analysed during the current study are available from the corresponding author on reasonable request.

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
