# Peer review of "A Cross-Sectional Study of Ocular Changes in Children and Adolescents with Diabetes Mellitus in Selected Health Facilities in Ghana"

_ijerph, 2022, doi:10.3390/ijerph19095295_

Round 1

Reviewer 1 Report

This is an interesting paper, I´ve enjoyed reading and reviewing it. The manuscript is fairly well written, although there are a fair number of editorial errors throughout. I agree that it would be an important topic to cover and that a study of this topic would be very helpful in immersing all the information about ocular changes in children and adolescents with diabetes mellitus in sub-Saharan Africa.

Minor editorial errors:

Line 187: in our protocol manuscript.22 à in our protocol manuscript [22].

Line 192: BMI was 20.7 (4.9) kg/m3à BMI was 20.7 (4.9) kg/m2.

Line 224: cortical (19, (32.8) types à cortical (19, 32.8) types.

Line 331: non-proliferative diabetic retinopathy (NPDR).--> NPDR.

Line 414: The grant number is missing in the funding column.

References: After the 9th reference, there are many spaces after the number of each article; and the references cited in the text are too old. The only reference in 2021 is the author's own article.   

Author Response

REVIEWER 1:

Minor editorial errors:

Line 187: in our protocol manuscript.22 à in our protocol manuscript [22].

Response Line 187:

The style of numbering for the reference has been corrected.

Line 192: BMI was 20.7 (4.9) kg/m3à BMI was 20.7 (4.9) kg/m2.

Response Line 192:

Unit of measurement for BMI corrected.

Line 224: cortical (19, (32.8) types à cortical (19, 32.8) types.

Response Line 224:

Correction has been done as follows; cortical (19, 32.8%) types.

Line 331: non-proliferative diabetic retinopathy (NPDR).--> NPDR.

Response Line 331:

Corrections have been made to read “This was a moderate NPDR”.

Line 414: The grant number is missing in the funding column.

Response Line 414:

The grant number has been provided in the funding column.

References: After the 9th reference, there are many spaces after the number of each article; and the references cited in the text are too old. The only reference in 2021 is the author's own article.

Response for References issues:

Issues with references corrected.

Reviewer 2 Report

COMMENTS

TITLE: A cross-sectional study of ocular changes in children and adolescents with diabetes mellitus in selected health facilities in Ghana

  1. The writing style can be improved for greater clarity and to shorten the manuscript. Below are comments for improvement of the paper.
  2. Abstract
    • Please remove the numbers in the abstract e.g. (1) Background; (2) Methods
    • Improve writing style e.g. Background. It would be best to state the primary objective first. “The main objective of the study is to determine the prevalence of diabetic retinopathy (DR), other diabetes- related ocular changes (e.g. cataracts, corneal ulceration) and non-diabetic ocular disease in Ghanaian children and adolescents. The secondary objective is to evaluate the relationship between these conditions and age at diagnosis, current age, diabetes mellitus (DM) duration, and participant’s sex.”
    • In the abstract there is a statement, “This cohort was followed up in a longitudinal study to determine the natural history of the ocular changes which will be reported later.” How long was the cohort followed up? The sentence can be shorter “This cohort was followed up for 3-years for the natural course of the ocular changes.”
    • The results in the abstract does not mention the results of the secondary objective especially for the major findings e.g. “There was a significant positive association between duration of the DM and the risk of cataract (p= 027).”
    • The conclusion is over-reaching. Rather than a national eye examination, maybe the authors should recommend that all young persons with diabetes should be regularly undergo eye examination throughout all the clinics where they are following up.
  1. The introduction is adequate but maybe there is a need to mention the prevalence/incidence of diabetes among children in Ghana if possible, even from that obtained from the data of the International Diabetes federation (IDF) or WHO.
  2. Methodology
    • What is the computed sample size and the basis? Will this sample size be adequate for both the primary and secondary objectives? Although it is mentioned that consecutive enrolment was done, how certain are we that the small sample size of 58 participants from 2 centers is representative of the health facilities in Ghana, and thus, the validity of the results and conclusions.
    • Generally, the methodology is well written and described.
  3. Results
    • Please use the generic names of drugs e.g. “mixtard” should be written as Biphasic Human insulin.
    • Improve the Tables.
      1. Ensure that table titles are complete e.g. Table 1. Distribution of Subjects according to Presenting Visual Acuity/Impairment, N= 58.
      2. Remove the horizontal lines in the tables. The table entries should be left justified instead of centered for ease of reading.
      3. For Table 1b. Instead of putting – (blanks) to indicate that none of the subjects had that impairment, maybe the entry should be 0. If blank is used, then it seems that they were not tested.
      4. Were tests of normality done to ensure that mean (mean age or mean duration of DM) is the valid test for central tendency? This is especially to answer Objective #2.
      5. Before going to discussion, there should be a paragraph that summarizes the results.
  1. Discussion
    • In the discussion, the limitation of a small sample size, and the inclusion of only hospitals for recruitment should be acknowledged.
    • Discuss also if the “other ocular changes” which were seen in this cohort of young persons with DM, are also common among those without diabetes based on other researches in Ghana.

Author Response

REVIEWER 2:

  1. The writing style can be improved for greater clarity and to shorten the manuscript.

Response to 1:

The writing style has been improved as suggested.

  1. Abstract
    • Please remove the numbers in the abstract e.g. (1) Background; (2) Methods
    • Response:
    • Numbers in the abstract has been removed.
    • Improve writing style e.g. Background. It would be best to state the primary objective first. “The main objective of the study is to determine the prevalence of diabetic retinopathy (DR), other diabetes- related ocular changes (e.g. cataracts, corneal ulceration) and non-diabetic ocular disease in Ghanaian children and adolescents. The secondary objective is to evaluate the relationship between these conditions and age at diagnosis, current age, diabetes mellitus (DM) duration, and participant’s sex.”

Response:

Writing style improved with all suggestions duly acknowledged.

    • In the abstract there is a statement, “This cohort was followed up in a longitudinal study to determine the natural history of the ocular changes which will be reported later.” How long was the cohort followed up? The sentence can be shorter “This cohort was followed up for 3-years for the natural course of the ocular changes.”
    • Response:
    • All suggestions duly acknowledged.
    •  
    • The results in the abstract does not mention the results of the secondary objective especially for the major findings e.g. “There was a significant positive association between duration of the DM and the risk of cataract (p= 027).”
    • Response:
    • This has been corrected as suggested by reviewer 2.

    • The conclusion is over-reaching. Rather than a national eye examination, maybe the authors should recommend that all young persons with diabetes should be regularly undergo eye examination throughout all the clinics where they are following up.

 Response:

                         All suggestions for conclusion duly acknowledged.

  1. The introduction is adequate but maybe there is a need to mention the prevalence/incidence of diabetes among children in Ghana if possible, even from that obtained from the data of the International Diabetes federation (IDF) or WHO.

Response:

            This suggestion is duly acknowledged.

  1. Methodology
    • What is the computed sample size and the basis? Will this sample size be adequate for both the primary and secondary objectives? Although it is mentioned that consecutive enrolment was done, how certain are we that the small sample size of 58 participants from 2 centers is representative of the health facilities in Ghana, and thus, the validity of the results and conclusions.

Response:

The computed sample size and the basis for computing has been explained.

    • Generally, the methodology is well written and described.

Response:

This comment is duly acknowledged.

  1. Results
    • Please use the generic names of drugs e.g. “mixtard” should be written as Biphasic Human insulin.

Response:

            This suggestion is duly acknowledged.

    • Improve the Tables.

      1. Ensure that table titles are complete e.g. Table 1. Distribution of Subjects according to Presenting Visual Acuity/Impairment, N= 58.
      2. Remove the horizontal lines in the tables. The table entries should be left justified instead of centered for ease of reading.
      3. For Table 1b. Instead of putting – (blanks) to indicate that none of the subjects had that impairment, maybe the entry should be 0. If blank is used, then it seems that they were not tested.

Response:

All suggestions to improve the tables are duly acknowledged with changes made.

      1. Were tests of normality done to ensure that mean (mean age or mean duration of DM) is the valid test for central tendency? This is especially to answer Objective #2.

Response:

Test of normality was done for variables including age of participants, age at diagnosis, BMI and duration of DM using the Kolmogorov-Smirnov test with p-values less than 0.05 considered as not normally distributed.

      1. Before going to discussion, there should be a paragraph that summarizes the results.

Response:

A summary of the results has been provided.

  1. Discussion
    • In the discussion, the limitation of a small sample size, and the inclusion of only hospitals for recruitment should be acknowledged.

Response:

            This suggestion is duly acknowledged in the discussion.

    • Discuss also if the “other ocular changes” which were seen in this cohort of young persons with DM, are also common among those without diabetes based on other researches in Ghana.

Response:

            This suggested discussion is duly acknowledged under discussion.

Reviewer 3 Report

A cross-sectional study of ocular changes in children and ado- 2 lescents with diabetes mellitus in selected health facilities in 3 Ghana

We consider that this study is of interest to increase awareness about the Diabetes mellitus consequences in the ocular globe, we think that the paper needs revision. Study limitations and conclusions are well addressed.

Abstract

The abstract requires editing in the background section, since the sentence «To determine the prevalence of diabetic retinopathy (DR), other diabetes related ocular changes (e.g. cataracts, corneal ulceration) and non-diabetic ocular disease in Ghana children and adolescents.»  does not represente the aim of the study and an introductory sentence on the importance of the theme must be added.

The sentence « This study sought to determine the hospital prevalence of ocular disease in children and adolescents with diabetes in Ghana, including DR, other diabetes-related ocular changes (cataracts, corneal ulceration) and non-diabetic ocular disease.» is more self-explanatory.

The information «recruited from Korle Bu Teaching Hospital, Accra) and Cape- 29 Coast Teaching Hospital, Cape-Coast, Ghana» does not need to be in the abstract section.

Materials and methods

Line 120

Children and adolescents aged 4–19 years diagnosed with DM with chronic disease,

Should be: …diagnosed with DM and concomitant chronic disease,…

Line 148

«with  particular attention to corneal innervation»

How was this feature evaluated?

Line 187

«our protocol manuscript.22» rephrase and place reference between square brackets

Line 193

«Mixtard insulin» – add reference

Line 201

«with overall prevalence of VI in the better eye per person, of 3 (5.2%).»

Please rephrase

Line 221

«were significantly the most common types (p<0.001).»

Delete significantly

Line 230

«Follicles»

Lymphoid follicles hipertrophy

Lije 231

Vitreal syneresis

Line 267

«on the increase»

replace for increasing

Line 269

«provides a comprehensive data»

replace for:…provides the first comprehensive…

Line 395

Other ocular changes, including cataracts, blepharitis and refractive errors, were 395 commonly observed. – add comas

Hopefully you can use the comments; I am looking forward to read the revised edition.

Author Response

REVIEWER 3:

Comments and Suggestions for Authors

We consider that this study is of interest to increase awareness about the Diabetes mellitus consequences in the ocular globe, we think that the paper needs revision. Study limitations and conclusions are well addressed.

Abstract

The abstract requires editing in the background section, since the sentence «To determine the prevalence of diabetic retinopathy (DR), other diabetes related ocular changes (e.g. cataracts, corneal ulceration) and non-diabetic ocular disease in Ghana children and adolescents.»  does not represente the aim of the study and an introductory sentence on the importance of the theme must be added.

The sentence « This study sought to determine the hospital prevalence of ocular disease in children and adolescents with diabetes in Ghana, including DR, other diabetes-related ocular changes (cataracts, corneal ulceration) and non-diabetic ocular disease.» is more self-explanatory.

The information «recruited from Korle Bu Teaching Hospital, Accra) and Cape- 29 Coast Teaching Hospital, Cape-Coast, Ghana» does not need to be in the abstract section.

Response:

Abstract revised with reviewer’s suggestions accepted.

Materials and methods

Line 120

Children and adolescents aged 4–19 years diagnosed with DM with chronic disease,

Should be: …diagnosed with DM and concomitant chronic disease,…

Response line 120:

Revision has been made in this sentence.

Line 148

«with  particular attention to corneal innervation»

How was this feature evaluated?

Response line 148:

This has been explained under methods.

Line 187

«our protocol manuscript.22» rephrase and place reference between square brackets

Response 187:

This revision has been done. 

Line 201

«with overall prevalence of VI in the better eye per person, of 3 (5.2%).»

Please rephrase

Response line 201:

This sentence has been revised.

Line 221

«were significantly the most common types (p<0.001).»

Delete significantly

Response line 221:

Significantly has been deleted.

Line 230

«Follicles»

Lymphoid follicles hipertrophy

Response line 230:

This has been revised accordingly.

Lije 231

Vitreal syneresis

Response line 231:

This has been revised accordingly.

Line 267

«on the increase»

replace for increasing

Response line 267:

This revision has been made.

«provides a comprehensive data»

replace for:…provides the first comprehensive…

Response line 269:

This has been revised.

Line 395

Other ocular changes, including cataracts, blepharitis and refractive errors, were 395 commonly observed. – add comas

Response line 395:

Reviewer’s suggestions duly acknowledged.

Hopefully you can use the comments; I am looking forward to read the revised edition.

Response:

Very grateful for your revision

Round 2

Reviewer 3 Report

Line 45

Rephrase the sentence :« and presence of prominent corneal nerves was significantly associated with a young age at diagnosis (p=0.004). »

Line 59 ...«young people is on the increase worldwide»...,

replace on the increase for  «has been increasing»

Author Response

REVIEWER 3: SECOND COMMENTS:

Comment 1:

Line 45

Rephrase the sentence:« and presence of prominent corneal nerves was significantly associated with a young age at diagnosis (p=0.004). »

Response:

Line 45 has been rephrased for clarity. 

Participants’ age at diagnosis was significantly associated with the presence of prominent corneal nerves (p=0.004).

Comment 2:

Line 59 ...«young people is on the increase worldwide»...,

replace on the increase for  «has been increasing»

Response:

The suggestion for Line 59 has been effected.

Traditionally, type 1 DM (T1DM) is more common in young people; however, type 2 DM (T2DM) in young people has also been increasing worldwide.
